# How does it feel to run in minimalist and advanced footwear technology shoes: A qualitative study involving male recreational runners

Kim Hébert-Losier[1,2]*, Hannah Knighton[1], Steven Finlayson[1], Benjamin Peterson[3,4]

**1** School of Sport and Human Movement, Division of Health, University of Waikato, Tauranga, New Zealand, **2** Research & Development, The Running Clinic™, Lac-Beauport, Canada, **3** Department of Podiatry, School of Health, Medical and Applied Sciences, Central Queensland University, Rockhampton North, Australia, **4** S.P.O.R.T Research Cluster, Central Queensland University, Rockhampton North, Australia

* kim.hebert-losier@waikato.ac.nz

## Abstract

We examined the perceptions and experiences of male recreational runners when using minimalist racing flats (FLAT, Saucony Endorphin Racer 2) and advanced footwear technology (AFT, Nike Vaporfly 4%) shoes, compared with their habitual shoes (OWN). Eighteen runners completed three 1.5 km outdoor trials, running in OWN first, followed by FLAT and AFT in a randomised counter-balanced order. Semi-structured interviews conducted before and after each trial provided qualitative data, analysed using a six-phase reflexive thematic approach. Five interconnected themes emerged: 'novelty and familiarity', 'feel', 'performance', 'biomechanics', and 'injury'. Runners' perceptions were shaped by iterative feedback loops combining experiential, educated, and instinctual assessments. OWN shoes were generally ranked highest for comfort and lowest for perceived injury risk due to familiarity and balanced design. Novel shoes elicited mixed reactions. FLAT shoes were valued for their lightweight and natural feel, but raised concerns about discomfort and potential injury from minimal cushioning and support. AFT shoes were appreciated for their bounciness and performance potential, but raised concerns about instability and excessive cushioning. Findings indicate that footwear comfort is multifaceted, context-dependent, and not always aligned with performance or injury reduction. Runners often prioritised performance over comfort in competitive scenarios, highlighting the inherent trade-offs in footwear selection and the importance of personalised approaches. Concerns about injury and biomechanics changes underscore the need for gradual transitions to novel footwear. By adopting a real-world approach, this study advances understanding of footwear perceptions, emphasises the dynamic and subjective nature of runners' experiences, and offers practical implications for runners, clinicians, and shoe manufacturers.

**Data availability statement:** The data associated with the development of the themes discussed in this article are presented throughout the text as direct quotes from the participating runners. The transcribed datasets are not publicly available as the research participants did not provide consent for their data to be shared in online public repositories and the Ethics Committee deemed it would be ethically inappropriate to share the data openly. Data requests can be sent to the Human Research Ethics Committee of the University of Waikato (humanethics@waikato.ac.nz) citing the approval number HREC(Health)2020#83. More detail on the Human Research Ethics Committee can be found at this link: https://www.waikato.ac.nz/research/research-enterprise/ethics/human-research-ethics-committee/.

**Funding:** This work was internally funded by Te Huataki Waiora School of Health, University of Waikato, New Zealand, were used to purchase all footwear used as part of this research (grant or award number: not applicable). This work was not endorsed by any footwear company.

**Competing interests:** Kim Hébert-Losier is a speaker for the Running Clinic, a continuing education organization that translates scientific evidence to healthcare professionals and the public. Internal university funds from Te Huataki Waiora School of Health, University of Waikato, New Zealand, were used to purchase all footwear used as part of this research (grant or award number: not applicable). This work was not endorsed by any footwear company. This does not alter our adherence to PLOS ONE policies on sharing data and materials.

## Introduction

Running is a popular activity with many health benefits [1], with most participants running recreationally [2]. Selecting the 'right' running shoe can enhance comfort [3,4], increase running pleasure [5], and potentially mitigate injury risk [6]. The running shoe market now offers a range of options, from minimalist designs [7] to more maximalist shoes with advanced footwear technology (AFT) [8].

Minimalist shoes aim to interfere minimally with natural foot motion [9] and are often considered more 'natural' with some believing the body is 'designed to be without trainers' [10]. Minimalist shoes offer a potential performance advantage as their lighter mass may reduce the metabolic cost of running [11], and have been suggested to help reduce the risk of running-related injuries [7], namely by reducing knee loads [12]. However, their design increases foot and ankle loads [13] and a reduction in injury incidence remains empirically unsupported [14].

In contrast, experts propose AFT defines a "performance-enhancing footwear technology that combines lightweight, resilient midsole foams with rigid moderators and pronounced rocker profiles in the sole." [8]. Although there is no true consensus definition for AFT shoes [15], they typically feature a thick midsole and stiff curved element that increases longitudinal bending stiffness [15], with their overall design aimed at reducing running energy costs and enhancing performance. Anecdotally, they are described as 'bouncy', with runners claiming 'never experiencing something like them' [16]. However, their rapid adoption raises injury concerns [15,17], as features like increased stack height may compromise stability [18], potentially leading to injuries such as ankle sprains especially during sharp turns in races [19].

Despite their novel features, AFT shoes are more similar in design to traditional and maximal running shoes than minimalist ones, offering cushioning that runners often find more comfortable [20,21]. Familiarity and cushioning are key factors influencing shoe comfort [22], and purchasing decisions [23]. However, comfort in novel shoes is rarely studied outside of laboratory settings, which may not reflect real-world experiences. Although objective measures of comfort such as visual analogue scales [24] and preference rankings [24] exist, comfort remains a 'feeling of a human' [25] and a multifaceted experience [3].

We aimed to explore recreational runners' perceptions of novel shoes in a real-world setting. Specifically, we examined the perceptions and experiences of male recreational runners running outdoors with AFT (Nike Vaporfly 4%) and lightweight minimalist (Saucony Endorphin Racer 2) shoes, using their habitual shoes as a baseline comparator.

## Materials and methods

### Participants

We recruited male recreational runners who were injury-free for at least three months, ran regularly (minimum once per week) for at least six months, had a 5 km personal best time between 20–30 minutes within the past year [5,20], and fit the available shoe sizes. These runners were recruited primarily via word-of-month,

snowball sampling, and personal contacts, and approached face-to-face, via e-mail, and through social media. Male runners were recruited due to the shoe availability at the time of the study and for sampling convenience. All those who expressed interest in the study completed the study. The first participant was tested 22 May 2021 and final participant 16 Nov 2021 (i.e., over 6 months). Eighteen runners consented to participate and completed the protocol (Table 1). The Human Research Ethics Committee [HREC(Health)2020#83] of the University of Waikato approved the experiment, which adhered to the Declaration of Helsinki. Participants were advised of the aims of the study (i.e., exploring their experiences running in different shoes), and had built a rapport with the main interviewer (SF) prior to agreeing to participate, who acknowledged working with the team on this topic and was interested in the results. All participants signed an informed consent document prior to participating, which assured their data would remain confidential. A research assistant (CM) was present to assist in the conduct of the study, but not in vicinity of runners during the one-on-one interviews.

## Design

This mixed-methods study involved a sequence of individual semi-structured interviews embedded within a randomised crossover trial. The study is primarily grounded in interpretive phenomenology, incorporating elements of ethnography. As reported elsewhere [21], participants attended one 90-minute session involving three 1.5 km run outdoors in vicinity of the University of Waikato Adams Centre for High Performance, Tauranga, New Zealand, on concrete in three different shoe conditions: OWN, habitual running shoe; FLAT, Saucony Endorphin Racer 2 racing flat minimalist shoes; and AFT, Nike Vaporfly 4% with advanced footwear technology (Table 1). Participants provided feedback on each shoe before and after every run, followed by a final interview to assess overall impressions, embodying elements of an ethnographic approach as participants were interviewed during and immediately after engaging with the shoes, and running in a real-world setting. The three-phase questioning process (Table 2) was iteratively and recursively refined in consultation

**Table 1. Characteristics of the 18 male recreational runners and shoes worn by participants during the study.**

| Runners | Characteristics | | |
|---|---|---|---|
| Age (y) | 31.2±10.5 | | |
| Height (cm) | 180.2±6.0 | | |
| Mass (kg) | 81.6±10.0 | | |
| Running experience (years) | 11.2±8.1 | | |
| 5 km best time in last year (min) | 23.1±2.1 | | |
| Weekly training (km) | 20.0±12 | | |
| Own shoe size (US men sizing) | 10.6±1.0 | | |
| **Shoes‡** | **OWN** | **FLAT** | **AFT** |
| Mass (g) | 308±42 [F,A] | 153±8 [O,A] | 211±12 [O,F] |
| Stack height (mm) | 24.6±7.2 [F] | 13.0±0 [O,A] | 31.0±0 [F] |
| Heel-to-toe drop (mm) | 11.2±5.7 [F] | 1.0±0 [O,A] | 7.0±0 [F] |
| Minimalist index (%)† | 28±15 [F] | 88±0 [O,A] | 48±0 [F] |
| Price (NZD) | 156±49 [A] | 190±0 | 380±0 [O,F] |
| **Running speed§** | **OWN** | **FLAT** | **AFT** |
| 1.1 km (m/s) | 3.57±0.30 | 3.62±0.35 | 3.65±0.40 |
| 400 m (m/s) | 4.18±0.35 | 4.09±0.36 | 4.15±0.38 |

*Notes.* Data are mean±standard deviation. AFT, advanced footwear technology (Nike Vaporfly 4%). FLAT, minimal road racing flat (Saucony Endorphin Racer 2). OWN, runners own habitual running shoes. ‡Data from right shoes only (size: US men 8.5 to 12). †Index range: 0% (lowest) to 100% (highest) degree of minimalism. §Runners ran 1.1 km at a self-selected comfortable pace and 400 m at a self-selected tempo run or 5 km race effort. No significant differences in 1.1 km ($P=0.447$) and 400 m ($P=0.071$) speed between shoes based on repeated measures analysis of variance. [O,F,A] Significant difference during post-hoc paired *t*-test comparisons ($P\leq0.05$) after repeated measures analysis of variance vs OWN, FLAT, and AFT, respectively.

**Table 2. Questions asked immediately before and after running with novel shoes at two different speeds for a total of 1.5 km.**

| Phase | Question |
|---|---|
| **1. The initial inspection**<br>Before running in each novel shoe | Q1. What stands out to you when you look hold and feel these shoes? |
| | Q2. Imagine you're in a shoe store trying them out. How do they feel? |
| | Q3. Would you buy these shoes? |
| **2. The running experience**<br>After running in each novel shoe | Q1. What did it feel like running in these shoes? |
| | Q2. How did running in these shoes feel compared to your own shoes? |
| | Q3. How did these shoes influence your running style? |
| | Q4. Did these shoes feel different at the slower and faster speeds? |
| | Q5. Use three words to describe how you felt running in these shoes. |
| | Q6. Did you enjoy running in these shoes? |
| | Q7. Would you buy these shoes now? |
| **3. The final ranking**<br>After running in all three shoes (own and both novel shoes) | Q1. Why did you rank them this way?<br>(*Runners were asked to rank the three shoes from most to least comfortable separately for the slower and faster speed*) |
| | Q2. Why did you rank them this way?<br>(*Runners were asked to rank the three shoes from best to worst performance, i.e., race situation*) |
| | Q3. Why did you rank them this way?<br>(*Runners were asked to rank the three shoes from lowest to highest perceived injury risk*) |
| | Q4. Do you know what brand or type of shoe these are?<br>(*Question asked to ascertain runners were blinded to the make, model, and type of the two novel shoes*) |

with experienced qualitative researchers (JB, CR) and recreational runners before study implementation. Given the in-situ and mixed-methods nature of the study, interviews were designed to be brief and used to capture immediate, experience-based reflections rather than prolonged discussions. This approach allowed for raw, first-hand impressions, minimising retrospective bias from participants.

This study primarily follows an interpretivist epistemology with a constructivist ontology, recognising that runners' perceptions of footwear are shaped by individual experiences. However, data collection was conducted using a neutral, structured interview approach inspired by neo-positivist methods to minimise interviewer bias. Qualitative data analysis was conducted using an interpretive phenomenological approach, aligning with the constructivist assumption that meaning is co-constructed through lived experience.

## Protocol

After obtaining written informed consent, participants and shoe characteristics were recorded (Table 1) using established procedures [9,21]. Experimental shoes were spray-painted black to conceal their brand and model.

Running trials were conducted on a flat 740 m concrete loop outdoors. Participants first ran 1.1 km at a self-selected comfortable pace sustainable for 30 minutes (~1.5 loops). After a 30 s standing rest, participants ran 400 m at a faster, race-like pace to simulate a tempo run or 5 km race effort. The 1.1 km distance aligned with the Running Shoe Comfort Assessment Tool [26], while the faster pace was included as the AFT and FLAT are designed for racing. Participants ran in OWN first as baseline, followed by AFT and FLAT in a random order. Participants wore a Garmin 245 Music watch (Garmin Ltd., Olathe, Kansas) during their running efforts (without listening to music) to monitor their 1.1 km and 400 m running times, enabling us to calculate running speeds (Table 1). The average atmospheric temperature and humidity during testing sessions was 12.8±3.3°C and 82.5±4.5%, with an average wind velocity of 15.1±3.0 km/h.

## Data collection

Participants had one-on-one interviews of one-to-three minutes in duration immediately before and after the running trials to discuss their perceptions regarding each shoe, limiting recall bias [27]. Prompting questions such as 'why?' or 'what do you mean?' were used to clarify responses, where required. The same male researcher with a research Masters who was a sport scientist, runner, and laboratory technician conducted all interviews (SF) using a 'neo-positivist' neutral-questioning approach, in which the interviewer remained neutral to participants' responses to mitigate social desirability [28], alongside developing a rapport prior to study commencement to enhance trust. This researcher was trained in interviewing by a qualitative expert (JB) prior to study commencement. To reduce social desirability bias, the interviews were conducted in a one-on-one setting and participants were assured their answers would remain confidential [29]. Interviews were voice recorded in real-time to enhance accuracy and limit recall bias [27]. Notes were taken alongside the interviews, especially to identify when questions were accidentally missed or not recorded. Voice-recorded interviews were transcribed *verbatim* using otter.ai. The same researcher (SF) verified the accuracy of the automated transcription process against the original recordings without these being returned to participants. As the interviews were conducted immediately before and after running in the novel shoes, returning the transcribed interviews would be susceptible to recall bias due to the extended timeframe [27].

## Data processing and analysis

All the data and transcripts available were used to process, analyse, and interpret data to achieve full saturation of the data. We used a six-phase thematic analysis process [30]. Familiarisation with the data begun during transcript formatting and continued with in-depth reading and note-taking (BP). The analysis followed an interpretive phenomenological framework, using a recursive approach to refine themes iteratively and accurately represent the data.

Initial codes captured runners' perspective on shoe comfort, performance, and injury. Codes and supporting quotes were organised in Excel and Word (BP, HK) using an inductive open coding approach, and initial themes were developed by identifying commonalities and recurring patterns among codes. Any discrepancies in interpretation were discussed between the two coders (BP, HK) until a consensus was reached or required a third viewpoint for reconciliation (KHL). Data were coded using a systematic process, where statements from runners were analysed and categorised into themes representing the phenomenon of interests [31,32]. Themes were refined through iterative review until coherent patterns emerged and evolved over time via a recursive process as the researchers engaged with and discussed the data. As participants did not directly validate the findings due to time constraints, participant availability, and to limit recall bias; the research team engaged in iterative theme validation to ensure alignment with participant narratives and inter-coder checks, minimising researcher interpretation bias of underlying concepts that were implied in participant responses. Themes and codes were reviewed by three members of the research team (BP, KHL, HK) at regular time points in a collaborative process to ensure accuracy, with participant quotes (abbreviated 'P') providing evidence and context. These researchers agreed on the final theme names and definitions, but the process remained iterative and flexible as all researchers engaged further with the data. The research team were all experienced runners and sport scientists, of whom two were medical professionals, two held doctorate research degrees, one was a professional triathlete, and one had qualitative analysis expertise. As experienced runners, medical professionals, and sport scientists, we (the research team) were aware of potential biases. To mitigate this bias, we critically examined how our assumptions shaped the coding and analysis process, engaging in reflexive discussions about our assumptions and potential biases. Participant quotes were maintained during the coding process to document evolving interpretations and ensure that themes were derived from the data rather than researcher preconceptions. Figures were generated to summarise findings, and content analysis (e.g., counts and percentages) was performed to complement the thematic analysis. In reporting the data, the Consolidated criteria for Reporting Qualitative research (COREQ) checklist [33] was followed.

## Results

### The emerging themes

Runners' perceptions of the novel shoes during the three-phase process were shaped through an experiential, educated, and instinctual lens, evolving iteratively as they gained new insights within this feedback loop (Fig 1). Five broad themes emerged from the analysis, detailed in Table 3. These overarching themes were highly interconnected, consistently appearing across the three research phases, and encompassing both positive and negative perspectives.

### Phase 1: The initial inspection

**What stands out?.** Runners acknowledged the low mass of both novel shoes, aligning with their significantly lower measured mass compared to their OWN shoes (Table 1).

*P17: The shoe, their pretty light (AFT). Overall, really lightweight (FLAT).*

Shoe construction was also a key standout. FLAT was perceived as highly flexible, lacking support, and offering minimal cushioning. Despite these limitations, its minimalism and lower mass were seen as advantageous and suitable for short, fast runs.

*P12: Probably not as supportive as my normal running shoes, but probably good for short faster racing.*

In contrast, AFT was viewed as stiffer, thicker, and 'chunky, chunky, chunky' (P16), with the midsole thickness suggesting 'a lot more cushion' (P1). Yet, others felt the cushioning fell short of expectations. For example, P3 noted 'Little to no

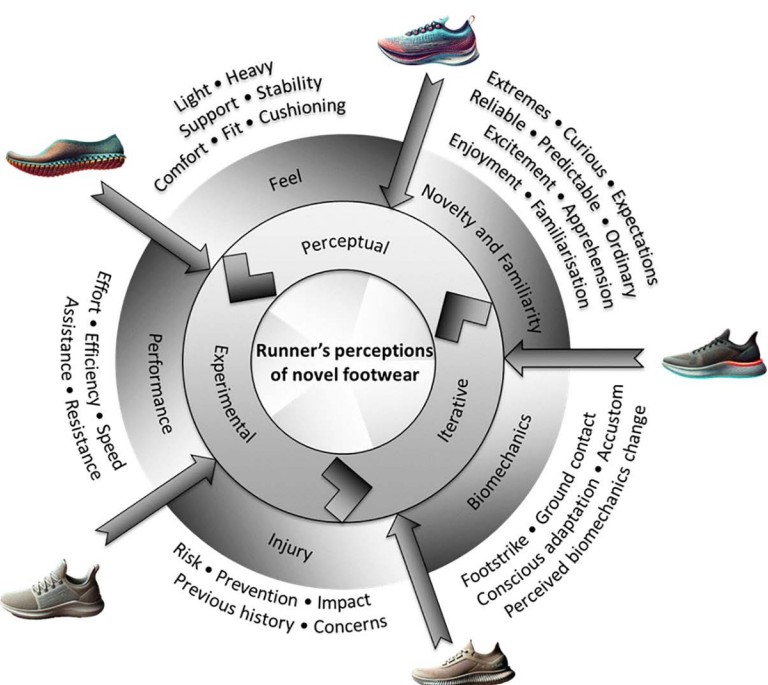

**Fig 1. The overarching themes and how they interact in describing the novel shoe experience of participating runners.**

 

**Table 3. Names and descriptions of the five main themes emerging from the semi-structured interviews conducted with male recreational runners (*n* = 18) derived from the three phases of this project: initial inspection, running experience, and final shoe ranking.**

| Theme | Description |
| --- | --- |
| Novelty and familiarity | Runners attributing a sense of familiarity or novelty (i.e., lack of familiarity) to shoes based on prior experiences, both perceived as either positive or negative. For example, a familiar shoe could be deemed 'reliable', but also 'mundane' or 'boring'. Novelty of shoes could elicit 'excitement' and 'curiosity', but also 'apprehension' especially in presence of extreme shoe designs. |
| Feel | Runners describing their expectations or perceptions of their own subjective feelings (e.g., comfort, fit, effort, and enjoyment) and shoe properties (e.g., cushioning, support, stability, mass, stiffness, responsiveness, resistance, compliance, and cushioning). The presence, excess, or lack of these subjective feelings or shoe properties could be perceived as either positive or negative. For example, a shoe could be perceived as having just the 'right' amount, too much, or too little cushioning. |
| Performance | Runners referring to their perceptions on whether shoes would improve or hinder their performances by making running feel faster or slower, less or more effortful, less or more efficient, or assisted or resisted. Runners perceived various shoe characteristics as positively or negatively affecting performance and their running pattern. For example, a shoe perceived as lightweight felt faster to some runners, but slower to others as perceived as 'they're doing nothing'. |
| Biomechanics | Runners addressing biomechanical changes to their running style, pattern, or gait associated to wearing novel shoes. Within this theme, there were two broad codes. The first code was the 'perceived biomechanical effect', which included perceptions of the shoes and their design influencing their running gait (e.g., propelling them forward or causing them to bear weight more laterally). The second code was the 'conscious adaptation' of runners, which was the perceived changes to their running gait to run more comfortably, efficiently, or safely in a novel shoe (e.g., avoiding heel striking to limit 'jarring' impact forces). |
| Injury | Runners discussing running-related injury aspects in relation to running in novel shoes, primarily in the context of increasing or decreasing their risk of future injuries. These impressions encapsulated shoe features, subjective perceptions of the shoes, deviations from their 'normal' running style, and suitability to their habitual running training. This theme encapsulated mostly the perceptions of runners on potential injurious implications of transitioning to novel shoes that were not suited to their body, running purpose or goals or habits, or previous injuries. These perspectives appeared more emotive and to be based on instincts and prior experiences, in particular, prior injuries. |

cushioning at the front and a very little bit of cushioning in the heel' and P12 stated 'Around the heel doesn't really have any padding or cushioning, like my regular shoes do.'

**How do they feel?.** Before running in the shoes, few runners considered how specific designs might affect their feel and running biomechanics. For instance, P18 remarked on the AFT: 'I would think that yeah, the very inflexible when it feels weird to run in it… I don't know what the heel-to-toe action would actually be like. Feels like I would just be slapping down one solid base'.

**Would you buy these shoes?.** At this early stage, most runners were hesitant about purchasing the novel shoes, with many emphasising the need to try running in them first. About a third of runners indicated they would consider buying the novel shoes (Fig 2), even 'without having run in them' (P3).

However, most responded negatively, citing the need for more information on the novel shoes' purpose, design, cost, feel during running, and suitability for their needs. Some runners felt FLAT was appropriate only for specific training sessions or for high-performing runners, a category they did not identify with.

*P8: If it was more of a like a track competition, quite possibly. But if it's for more of a just everyday out jogging, probably not from the initial feeling of it.*

Consistent with the most common reasons for buying their OWN shoes (S1 Fig); lack of comfort, perceived poor fit, and cost were perceived barriers to purchase. These concerns were reflected in runners' comments about FLAT:

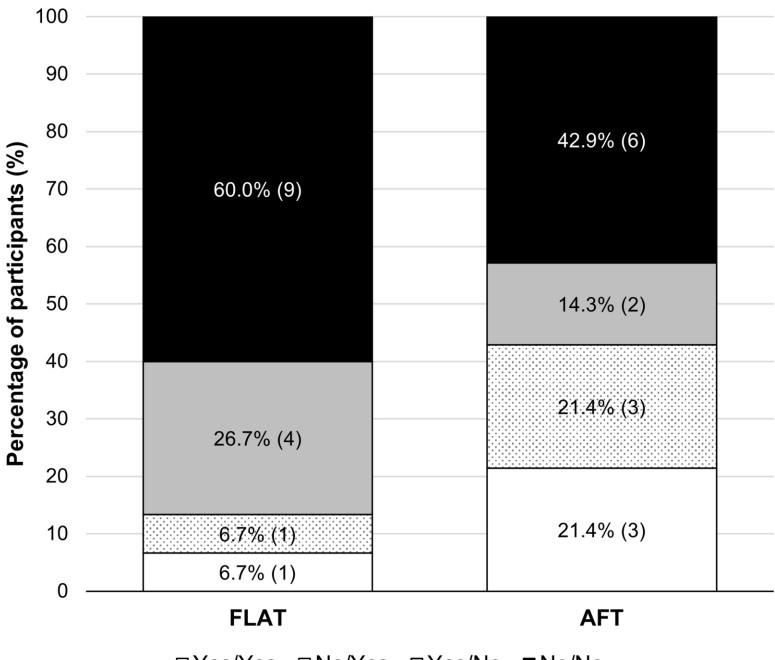

**Fig 2. Stacked bar charts of the responses from runners (*n* = 15 FLAT, *n* = 14 AFT) to whether they would purchase the novel shoes before and after running in them presented as before/after response.** The number of participants who provided a given response is in parentheses. Abbreviations: AFT, advanced footwear technology (Nike Vaporfly 4%). FLAT, minimal road racing flat (Saucony Endorphin Racer 2).

*P10: They don't feel comfortable.*

*P6: Probably not, they are not wide enough.*

*P13: It depends on how much they cost.*

Some runners hesitated to purchase due to unfamiliarity with the novel shoes, often perceived as 'too different' from what they were accustomed to, as stated P18 about AFT: 'They feel a little bit too different to what I normally run it. A little too bouncy.' Recommendations from peers would ease their concerns.

*P12: Would I buy them? Not without going through a bit more of a run. They're quite different to what I normally feel, so if I could go for a decent run try them out, maybe? Probably not straight off the store shelf, like this no. Maybe if it had a recommendation or something from a friend who said they're really good.*

In this early stage of shoe inspection, only one runner raised injury concerns about running in FLAT (P18), and another for AFT.

*P17: No. I don't feel like I could go for a long run in these (AFT), they would injure me straight away.*

**Phase 2: The running experience**

**Use three words to describe how you felt running in these shoes.** After running, participants most often described FLAT as light (44.4% of participants) and fast or quick (38.9%). AFT was most often described as bouncy (44.4%), and fast or quick (33.3%), as shown in Fig 3.

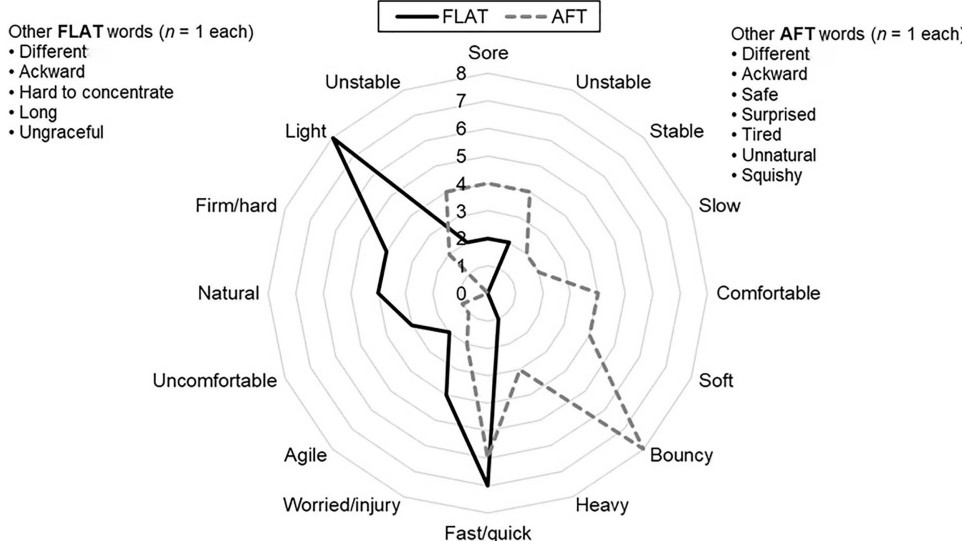

**Fig 3. Representation of the three words used to describe the novel shoes after running by runners (*n* = 18).** The radar plot represents the frequency of word use. Synonyms were counted as the same word. Abbreviations: AFT, advanced footwear technology (Nike Vaporfly 4%). FLAT, minimal road racing flat (Saucony Endorphin Racer 2).

**How did it feel to run in these shoes and compared with your own shoes?.** After running in the novel shoes, runners reported mixed views on the 'feel', comfort, mass, geometry, stability, and cushioning of both FLAT and AFT. The sole thinness of FLAT created a 'bare feet' feel with low cushioning and greater impact sensation. Some runners found its unstructured design unsupportive, uncomfortable, and harder to run in, especially over longer distances. However, others appreciated its lightness and responsiveness, which made running feel less effortful.

*P13: …quite light them being as light as they were. And, yeah, it's harder running like especially on hard pavement, they were less sole and harder sole probably than mine.*

*P17: Lighter, felt a bit faster, not as comfortable probably not as stable. It's better, not better, but faster.*

In AFT, runners had mixed reactions to the thicker, structured midsole. Some enjoyed its responsive, 'bouncy' feel, while others found it overly soft and cushioned. The responsiveness was described as providing 'more spring' (P10) by some, but a sense of excessive softness made the ride damp and unstable to others, 'like running on sand' (P3).

Runners reported biomechanical adjustments in both conditions. In FLAT, some runners felt the lack of cushioning and structure led to an 'uncoordinated' (P5) or 'jarring' (P14) run, requiring biomechanical alterations. Others felt the reduced material in FLAT provided a 'much more natural' (P3) experience, making running feel easier than in AFT. While some anticipated becoming accustomed or habituated to FLAT over time; others worried about injury with continued use.

P2: It *didn't feel as much cushioning, so I felt like I was plodding a little bit, but I feel like I could (get) used to that pretty quick.*

P4: *I think overall, if I were to have them longer term, I'd probably give myself (shin) splints or something like that.*

Injury concerns extended to the AFT condition, where biomechanical alterations were also seen as potentially harmful. Runners reported feeling unstable or 'rolly' (P1) in AFT, with some expressing being 'scared of corners' (P1) and worried about acute ankle or long-term knee injuries.

*P5: Because it's so cushioned turning corners felt like you were dicing with death. Like you're gonna roll your ankle.*

*P4: I feel like if I had them long term, I would have done a knee or something.*

**How did these shoes influence your running style?.** Runners described distinct and overlapping changes in their 'running style' for the two novel shoes, attributing these changes to either the 'perceived biomechanical effects' of the shoes or 'conscious adaptations' they made (Table 2).

Both shoes prompted a shift toward forefoot striking or faster forefoot weightbearing. In FLAT, runners consciously adapted their style to forefoot to reduce impact sensations, often described as 'jarring', 'shock', or 'shockwaves' (P1, P8). These sensations were linked to the shoes' reduced rearfoot cushioning and thickness compared to their habitual shoes, as P12 noted: 'It made me more conscious of how I'm placing my foot because normally I've got a thicker heel, and it made me feel like I was running on my toes more, on my forefoot'. Some runners rationalised that forefoot striking in FLAT would lessen their injury potential by lowering impact forces or excessive loads on previously injured tissues.

*P14: You had to concentrate hard on how you're landing or stepping to try and minimise the impact and try and minimise potential injury.*

In AFT, the shift toward forefoot striking or a quicker transition to forefoot was primarily attributed to the shoe's rocker profile, which propelled runners forward. P14 described it as 'that rocker-type feel on the shoe', and P11 noted they 'just sort of rolled into the front and sort of bounced off it a bit more'. Conscious adaptation was also required in AFT, with runners emphasising the need to maintain a straight line to avoid unpredictable lateral movements of the shoe.

*P8: (I was) trying not to turn much just staying in a straight line, just because of the way the heels react.*

Runners suggested a transition period could be beneficial to get accustomed and habituated to the novel shoes. Without one, P15 noted they could 'start to get a bit sore' if they 'ran wrong' in FLAT. Becoming accustomed with AFT reduced concerns about running style and injury risk, as P18 explained 'I feel like I stopped paying attention to it (running style) too much because I wasn't worried about getting injured because it felt so spongy on the bottom, so I was just kind of going, which was quite nice'. Alterations in running styles were often attributed to the novelty/unfamiliarity of the shoes and contrasted with the familiarity of their OWN habitual running shoes.

*P4: I think because they're newer (the FLAT), I felt like I was running on different parts of my feet.*

Running in the novel shoes was perceived to influence effort levels both positively and negatively. Positively, runners described feeling 'assistance', with the shoes propelling them forward, increasing or decreasing their stride length, and enabling their feet to move faster.

*P10: I felt like my strides are longer and I was able to run probably a bit more naturally (in AFT) than I would be in my normal shoes.*

*P16: I felt like I could get a maybe a longer stride, maybe, because they (the FLAT) were lighter.*

Conversely, some runners experienced 'resistance', where running felt more effortful. In FLAT, this resistance was attributed to a lack of support, while in AFT, it stemmed from excessive cushioning and softness dampening force application to the ground.

*P1: They (FLAT) are a bit more like it's just me and my running rather than the shoe helping with that movement.*

*P5: I had to concentrate a lot harder (in AFT)...you had to really push off each stride to keep pace, because it just feels like you're running in sand.*

**Did the shoes feel different at faster and slower running speeds?**

A subset of runners perceived little or no difference between speeds, with P3 stating FLAT felt 'very much the same' and P11 noting AFT was 'not noticeably' different and 'overall felt good at both paces'.

However, about half of runners noticed differences in FLAT between speeds. Only P18 preferred FLAT at the slower speed, explaining they '…just felt more impact through my legs at the fastest speed compared to my own shoes' and '…it was kind of hammering a bit more'. Most preferred FLAT at the faster speed, especially alongside biomechanical changes to lessen the 'jarring' (P14). Running faster, runners perceived lower contact times, improved foot-ground interactions, and quicker feet movements, enhancing their efficiency and comfort, and reducing their perceived effort.

*P2: Yes, slightly when I going faster it was good because it could move my feet a bit faster, the lighter it was, it felt a bit easier.*

*P14: At higher speeds when you're concentrating a bit more and I guess your striding out it felt better…*

Most runners noted differences in the feel of AFT between speeds, with preferences divided, but commonly linked to perceived comfort and ease of running. Those favouring faster speeds found AFT 'more comfortable' (P8, P16), easier, and more stable to run in, with 'less control' (P17) running slower. P5 found AFT 'only marginally better at faster speeds', but only 'on a straight, definitely not on a corner' due to feeling unstable.

In contrast, some struggled with responsiveness and control at the faster pace. P3 found it hard to 'get to a good pace' and felt the shoes lacked responsiveness, noting 'I was squidging down as I made each step'. Similarly, P4 described a sense of heaviness, saying 'Yeah, they felt heavier anyway'.

**Did you enjoy running in these shoes?.** For both FLAT and AFT, runners highlighted themes of 'feel', 'purpose', 'novelty', 'performance', and 'injury risk' as influencing their running enjoyment. The novelty of the shoes evoked mixed reactions. Some runners appreciated their uniqueness, describing FLAT as a 'good change' (P4) and AFT as 'a really novel feeling' (P4) or 'interesting' (P11). Others found the differences off-putting, with FLAT perceived as offering 'no support' (P7) and AFT requiring 'more effort to maintain that comfortable pace' (P3).

Comfort played a significant role in enjoyment. In FLAT, the thin sole and perceived lack of support compromised comfort for some runners, with P6 noting 'the shape of my feet don't really suit the type of shoe' and P10 describing foot pain from the thin base. Others found FLAT surprisingly comfortable 'straight out of the box' (P16) or appreciated the connection to the ground, even if it felt less cushioned than their OWN shoes (P17).

Similarly, AFT elicited mixed responses. Some runners found it comfortable, 'nice and light to run', and free from 'hardship on my feet' (P18). Conversely, others found it 'too stiff' (P6), 'not feeling very nice on the foot' (P17), or 'unnatural' due to excessive cushioning and stiffness, which hindered efficiency and enjoyment. One runner considered AFT suitable for races, but less appealing for everyday runs due to concerns about injury and comfort.

*P17: If I was racing I would definitely wear these shoes. But if I want to go out and not get injured and have a more comfortable run I'd probably use mine.*

**Would you buy these shoes now?.** After running, five participants changed their minds for each shoe as to whether they would purchase them compared to after their initial inspection (Fig 2). These changed responses highlight how

runners' perspectives changed because of their iterative, experiential, and perceptual experiences linked with running that was unique to looking, holding, and 'trying on' the shoes. Purchasing decisions for both shoes hitched on shoe familiarity, 'feel', comfort, match to a runner, need for further information, and cost.

Only two runners expressed interest in purchasing the FLAT after running (one changing from 'would not buy' to 'buy', Fig 2), citing feelings of familiarity, reassurance, and 'because nothing bad happened' (P18). Even if enjoying the novelty, some runners declined due to unfamiliarity, limited suitability to their running needs, or mismatch with their athletic identity – being 'at the lower end of the scales' and not a 'performance runner' (P15). For marathon training, P12 found FLAT unsuitable, requiring 'a shoe with more support for those longer runs'. Lack of support was linked to 'lack of comfort' (P11), 'wouldn't be very pleasurable' (P7), and 'might actually cause more injury' (P1), all barriers to purchase. Specific features were deal-breakers for certain runners, like 'that thin sole, didn't really cut it for me' claimed P11. Though an adjustment period might help, many felt FLAT was 'too different' from their habitual shoes, with P5 admitting, 'I don't know how to run in them'.

Similarly, while runners were intrigued by AFT, many were deterred by its extreme design, described as 'a bit too much one way' compared to FLAT being 'a bit too much the other way'. This unfamiliarity and extreme design pushed runners out of their comfort zones, as P16 noted 'they're too different away from where I'm comfortable…'.

Uncertainty about the purpose or design of the novel shoes would lead some runners to seek further clarification. In absence of this additional information, though, decisions regarding purchase often relied on feel and perceived comfort. P18 remarked that AFT 'feel better than my own running shoes'.

*P14: Yes, but I would want to talk to someone about them, who can explain what's going on. If we're just basing it on the feel, then yes.*

### Phase 3: The final ranking

**Comfort at slow and fast speeds.** OWN was most often ranked as the most comfortable at slower speeds, while AFT was preferred at faster speeds (Fig 4). FLAT was consistently ranked as the least comfortable at both speeds. Some runners adjusted their rankings between speeds; for instance, P4 found AFT 'a bit heavy' at faster speeds and preferred it for slower runs. Conversely, P6 felt its thick sole made slower runs less comfortable. Overall, the familiarity of OWN shoes and their balance of cushioning, support, and design features contributed to positive comfort rankings.

*P5: My own just feel like they have more of what I needed, more cushioning, more support. They were stiff enough.*

FLAT ranked lowest in comfort due to its lack of familiarity, cushioning, and ease of running, with P14 confirming it as the worst as 'too light and not enough cushioning' and P5 adding it was 'too hard to run slow in, with no cushioning'.

Conversely, some runners preferred the novel shoes, with P16 stating they were 'both probably better than mine'. Ease of running enhanced comfort, with P2 claiming for AFT it 'felt like it was a lot easier to run with…because they were a lot more springy. You're not putting in as much energy at the same pace'.

**Performance.** In terms of performance ranking, FLAT and AFT were polarising. The two novel shoes exhibited the highest frequencies for both the most and least preferred shoes for overall performance, reflecting runners' strong likes or dislikes from a racing perspective.

*P2: …(AFT) feel the fastest for the least amount of effort.*

*P6: …(AFT) are (ranked) the lowest because they are too heavy and don't provide enough flexibility.*

Preferences for performance shoes were linked to familiarity, ease of running, and perceptions of shoe features. Lighter shoes were desirable for some, even at the cost of comfort.

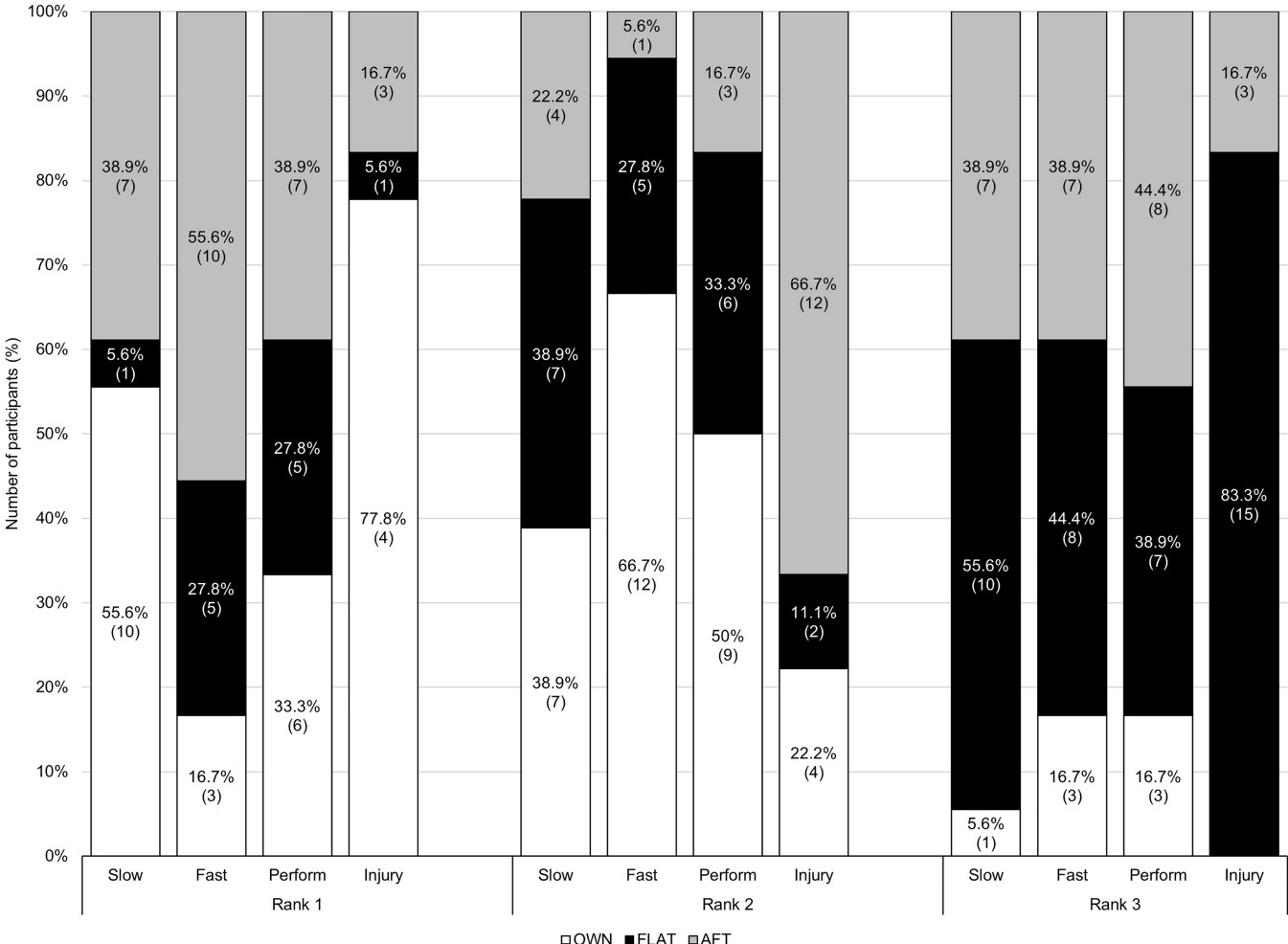

**Fig 4. Stacked bar charts of runners (*n* =18) preferences for different criteria following running a total of 1.5 km at two different speeds.** Rankings represent most preferred (rank 1) to least preferred (rank 3) for comfort at the slow speed, comfort at the fast speed, and overall performance. Rankings represent perceived lowest injury risk (rank 1) to highest injury risk (rank 3). Abbreviations: AFT, advanced footwear technology (Nike Vaporfly 4%). FLAT, minimal road racing flat (Saucony Endorphin Racer 2). OWN, runners habitual footwear.

*P4: I think in a race situation, I'd probably sacrifice a bit of comfort for a bit more lightness and performance.*

Opinions on perceived shoe mass and features, however. P4 found AFT heavy, but P18 found it light. FLAT was favoured by P3 for having 'some cushion but not too much', while P14 disliked it as 'too light, not enough cushioning'. A third of runners (Fig 4) preferred OWN for racing, citing familiarity, comfort, and stability.

*P2: I guess I am used to my own, and they're quite stable.*

*P8: I'd probably be more comfortable with that overall. I think I'd perform better because I'm used to the feeling*

**Injury risk.** Runners most often ranked OWN as the shoe with the lowest perceived injury risk (Fig 4), attributed mostly to familiarity and comfort. FLAT was most often ranked highest for injury risk due to its lack of cushioning and support.

*P7: Because I've run in my own shoes a lot, so I rank that one because I haven't really had an injury on them properly.*

*P8: …when running on any hard surfaces and everything, I'll just be getting a lot more jarring (in FLAT) through my joints and lower back and everything else so it's not very good for me.*

Some runners also expressed injury concerns with AFT, linked not only to unfamiliarity, but also to excessive cushioning and perceived instability on corners. P16 described it as 'too rolly' and P1 noted 'a little bit of slip', which raised concerns.

**Blinding.** Of the 18 runners, five correctly identified the brand of the AFT (27.8%) and two the brand of the FLAT (11.1%). No runner knew the specific model of these shoes.

## Discussion

Our findings provide novel insights into the subjective experiences of male recreational runners running in three distinct footwear types: AFT, minimalist, and habitual shoes (OWN). By adopting a qualitative approach, we captured rich, multi-faceted data and explored the complexity of runners' experiences and perceptions of comfort, performance, and injury risk running in varied footwear. To our knowledge, this is the first study to examine subjective perceptions of runners wearing AFT applying a qualitative lens, filling an important gap spanning running-related injury, biomechanics, and footwear design literature. Our findings also expand our understanding of minimalist footwear and how runners experience them [10], highlighting the contrasting extremes of shoe design.

The ecological validity of our study is a key strength. Conducting trials in an outdoor environment reflective of real-world training and racing ensured that participants' perceptions were examined in practical contexts. This consideration is important, as studies collectively indicate significant differences in running biomechanics between outdoor and treadmill running [34], with an overall greater enjoyment and lower perceived effort experienced when exercising in outdoor than indoor environments [35]. By including three phases in our shoe-appraisal process (i.e., initial inspection, running, final ranking, Table 2), two running speeds, and shoe wear over 1.5 km, we comprehensively examined how perceptions of running shoes evolve over time in a feedback loop that includes experiential, perceptual, and iterative processes, revealing the dynamic nature of footwear comfort, performance, and injury perceptions.

Our findings build and expand on the 'comfort filter' paradigm [36], which posited that runners select comfortable shoes using their comfort filter, automatically reducing their injury risk. Our results demonstrate that runners use a more advanced feedback loop in shoe appraisal than relying solely upon comfort, considering multiple input sources and fine-tuning their perspectives over time. Our data reinforce that comfort is a much more complex phenomenon and is not static, supporting that footwear comfort is multifaceted and subjective in nature [3].

Participants highlighted the need to modify their biomechanics in novel shoes [21], especially when shoes felt uncomfortable initially. These observations suggest that comfort/discomfort are stimuli that shape the running biomechanics and experiences of runners, with both immediate sensations and longer-term body-shoe interactions in turn affecting comfort and perceptions. This adaptation process, as well as the overt injury-related concerns some runners expressed running in the novel shoes, underscores the importance of a gradual transition to novel footwear to allow familiarisation and habituation. Bone stress injuries to the foot have been reported for runners transitioning to both AFT [17] and minimalist [37] shoes. In the present study, male recreational runners highlight how the underlying injury mechanisms might differ, with the excessive cushioning making runners feel they were 'in sand' or 'squidging' in AFT, but lack thereof increasing the 'jarring' and 'shock' sensations in FLAT. The concerns expressed from runners regarding ankle sprains in turns in AFT are noteworthy, pointing toward compromised frontal plane stability [19] and increased ankle eversion excursion [18] linked with their design. We are unable to confirm whether the perceived biomechanical changes runners felt running in novel shoes were in fact linked with actual biomechanical changes, which is important to consider in the context of existing evidence demonstrating low accuracy of runners' self-reported gait parameters, for example, foot strike patterns [38]. Future

research should seek to corroborate subjective perceptions of runners to biomechanical measures. Furthermore, it is challenging to unequivocally recommend specific transitioning timelines for novel footwear from existing literature [39], with a minimum of 4–8 weeks of gradual transitioning to minimalist shoes recommended for general musculoskeletal adaptations [39]. As AFT are more like traditional shoes in their design and comfort levels than minimalist shoes [21], the habituation period to AFT compared to minimalist shoes may be quicker for runners used to running in more traditional footwear. Nonetheless, the time and training requirements for habituation and adaptation to novel shoes remain relatively unknown and are likely specific to individuals and shoes.

Familiarity emerged as a critical factor in comfort, performance, and injury risk perceptions, with both familiarity and comfort reported to underpin footwear selection [22]. OWN shoes were consistently ranked highest for comfort at slower running speeds and lowest for injury risk due to their familiarity and perceived balanced design. In contrast, the novelty of AFT and FLAT evoked mixed reactions. Some runners appreciated the unique features of these shoes, while others struggled with their extremes, citing issues like instability, excessive stiffness, or insufficient cushioning. The data drawn from the interviews highlight the conflict between familiarity and novelty in our runners' preference, with familiarity shown elsewhere to evoke 'liking' and novelty to evoke 'interest' [40]. Runners tended to perceive familiar shoes (e.g., their own) more favourably likely due to an increased processing fluency – a phenomenon where familiar stimuli are easier to mentally process and therefore feel more comfortable or preferable [41]. At the same time, novelty seeking can drive preference due to its association with excitement, curiosity, and stimulation [42]. Noteworthy, runners' perspectives often shifted as they progressed through different stages—holding the shoes, trying the shoes on, running slowly, and running at faster speeds. These evolving perspectives suggest that running in new shoes provides a richer source of information than static or confined assessments, with comfort being influenced not just by the shoe's physical properties, but also by the context and intensity of use. Therefore, typical in-store processes might not be sufficient in facilitating informed decision making for runners seeking to purchase new shoes. The more positive perceptions towards novel shoes after participants ran with them reflect the mere-repeated-exposure paradigm, where repeated exposure enhances the positive affect of individuals towards a stimulus object [43].

Although some overlap existed, the shoe perceived to maximise comfort was not necessarily the same as the one considered optimal for performance advantages or mitigating injury risk. Many runners believed that lightweight shoes could offer a performance benefit, consistent with empirical evidence linking reduced shoe mass to lower metabolic cost during running [11]. Runners were often willing to prioritise performance over comfort in racing scenarios, challenging the notion that superior comfort inherently enhances running economy and – by proxy – performance [44,45]. This finding highlights a critical trade-off, as the most comfortable shoe may not be ideal for maximising performance. Consequently, runners must navigate competing priorities (comfort, performance, and injury prevention) when selecting footwear [6], making shoe selection a complex process.

Our study also reinforces that footwear perceptions are not only the byproduct of comfort, but also closely tied to other factors, such as fit, cushioning, stability, and support, as illustrated in Fig 1. While comfort is central to decision-making processes and footwear selection [46], it is not an isolated factor. Runners draw on multiple sources of information, including prior experiences, running sensations, and personal preference, to evaluate the suitability of a shoe. Notably, the same shoe can evoke vastly different experiences and enjoyment levels among individuals, highlighting the importance of personalised approaches to footwear selection and prescription.

Several limitations should be acknowledged. Our sample consisted solely of male recreational runners, limiting the generalisability of findings to female recreational runners and other running populations (e.g., professional athletes or different age groups). Additionally, we examined only one model of AFT and minimalist shoes, which may not represent the full spectrum of footwear designs available [9]. Furthermore, each trial involved a 1.5 km run, which may not reflect real experiences during long-distance running or daily training. Future studies should extend the running distance and expand sample diversity, exploring broader demographics and a wider range of shoe models and brands

to build on these findings. The trials were conducted outdoors, which confers high ecological validity, but lacks environmental standardisation. Variables such as ambient temperature, relative humidity, and wind velocity may have influenced runners' comfort perceptions. For instance, elevated temperatures can increase in-shoe heat and moisture, potentially affecting thermal comfort [47]. Ambient temperatures also influence footwear properties and material characteristics [48,49], which can in turn influence runners' comfort perceptions. Although all participants ran on the same surface and followed the same route, we acknowledge that environmental variability may have contributed to subjective comfort ratings. While an interpretive phenomenological framework (interpretivist epistemology) was used for data analysis, data collection followed a neo-positivist approach (constructivist ontology) to minimise researcher influence on participant responses. This hybrid methodology may have influenced the depth of interpretative engagement. Future research could benefit from a more constructivist interviewing approach, such as semi-structured or phenomenological interviews, to explore runners' subjective experiences in greater depth. Finally, as the research team consisted of experienced runners and sport scientists, it is important to acknowledge that our expertise and experiences may have shaped the analysis and interpretation process despite the steps taken to ensure the integrity of participants' perspectives.

Despite these limitations, this study makes an important contribution to the understanding of runners' footwear perceptions, particularly regarding AFT. By undertaking a qualitative study in an ecologically valid setting, we offer valuable insights into how runners interact with, and perceive, novel footwear in real-world contexts. These findings expand upon existing theoretical frameworks and have practical implications for runners, coaches, clinicians, and shoe manufacturers. By recognising the subjective and dynamic nature of male recreational runners' appraisal of novel footwear conditions, we can better inform footwear selection to enhance the running experience and overall satisfaction of runners.

## Conclusions

This study contributes to a deeper understanding of runners' footwear perceptions and offers practical insights relevant to shoe design and selection. The male recreational runners in our study expressed mixed perspectives of AFT and minimalist footwear, which were informed by prior experiences and moulded through an experiential and iterative process as they interacted with the shoes. Sensations of comfort, discomfort, cushioning, and stability; perceptions of performance and risk of injury; and impressions of biomechanical changes and conscious adaptations to novel footwear shaped runners' perceptions. The findings of this study support that footwear preferences are complex, and how footwear has the potential to influence runners' running experiences. The findings of this study support the need for a gradual habituation period to both AFT and minimalist footwear.

## Supporting information

**S1 Fig. Top three self-reported main reasons of participants (*n* = 18) for purchasing their current habitual running shoes.** The percentage (%) represents the number of participants who selected the reason as their first, second, or third reason from most (rank 1) to least (rank 3) important. The number of participants who selected the reason as their first, second, or third reason is provided in parentheses, respectively.
(DOCX)

## Acknowledgments

The authors would like to acknowledge the **runners** for their voluntary participation in this study. We also thank **Dr Julie Brice** and **Dr Codi Ramsey** for qualitative methodological advice, and **Dr Jean-Francois Esculier** for discussions during conceptualisation. We thank **Courtney Mitchel** for assistance during the data collection process.

## Author contributions

**Conceptualization:** Kim Hébert-Losier.

**Data curation:** Kim Hébert-Losier, Steven Finlayson, Benjamin Peterson.

**Formal analysis:** Kim Hébert-Losier, Hannah Knighton, Benjamin Peterson.

**Investigation:** Kim Hébert-Losier, Hannah Knighton, Steven Finlayson.

**Methodology:** Kim Hébert-Losier, Hannah Knighton, Steven Finlayson, Benjamin Peterson.

**Project administration:** Kim Hébert-Losier.

**Resources:** Kim Hébert-Losier.

**Supervision:** Kim Hébert-Losier.

**Visualization:** Kim Hébert-Losier, Hannah Knighton, Benjamin Peterson.

**Writing – original draft:** Kim Hébert-Losier, Hannah Knighton, Benjamin Peterson.

**Writing – review & editing:** Kim Hébert-Losier, Hannah Knighton, Steven Finlayson, Benjamin Peterson.

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
