## [Decision Letter · Decision Letter 0]

5 May 2025

Dear Dr. Hébert-Losier,

Thank you for submitting your manuscript to PLOS ONE. After careful consideration, we feel that it has merit but does not fully meet PLOS ONE’s publication criteria as it currently stands. Therefore, we invite you to submit a revised version of the manuscript that addresses the points raised during the review process.

We look forward to receiving your revised manuscript.

Kind regards,

Yaodong Gu

Academic Editor

PLOS ONE

Journal Requirements:

Kim Hébert-Losier is a speaker for the Running Clinic, a continuing education organization that translates scientific evidence to healthcare professionals and the public. Internal university research funds were used to purchase all footwear used as part of this research.

5. Please provide a complete Data Availability Statement in the submission form, ensuring you include all necessary access information or a reason for why you are unable to make your data freely accessible. If your research concerns only data provided within your submission, please write "All data are in the manuscript and/or supporting information files" as your Data Availability Statement.

6. Please amend your manuscript to include your abstract after the title page.

7. Please upload a copy of Figure 5, to which you refer in your text on page 21. If the figure is no longer to be included as part of the submission please remove all reference to it within the text.

8. Please remove all personal information, ensure that the data shared are in accordance with participant consent, and re-upload a fully anonymized data set.

9. We notice that your supplementary figures are uploaded with the file type 'Figure'. Please amend the file type to 'Supporting Information'. Please ensure that each Supporting Information file has a legend listed in the manuscript after the references list.

Reviewers' comments:

Reviewer's Responses to Questions

**Comments to the Author**

1. Is the manuscript technically sound, and do the data support the conclusions?

Reviewer #1: Yes

Reviewer #2: Yes

2. Has the statistical analysis been performed appropriately and rigorously?

Reviewer #1: Yes

Reviewer #2: Yes

3. Have the authors made all data underlying the findings in their manuscript fully available?

Reviewer #1: Yes

Reviewer #2: Yes

4. Is the manuscript presented in an intelligible fashion and written in standard English?

Reviewer #1: Yes

Reviewer #2: Yes

Reviewer #1: It is an interesting study, based on questionnaire method. I specifically admire the originality of this work, regarding the comfort level of shoes. I consider that sections Introduction and Discussions gets the reader familiarized with the topic.

Reviewer #2: This study conducted real-world outdoor experiments with 18 male recreational runners, comparing their subjective perceptions and running experiences with minimalist shoes and advanced footwear technology (AFT) shoes. It highlights the trade-offs runners make between comfort, performance, and injury risk when selecting footwear. The study contributes to a deeper understanding of runners' footwear perceptions and offers practical insights for shoe design and selection. However, its main limitations include a homogeneous sample, limited shoe models, and reliance on subjective data. The detailed reviewer comments are as follows:

1. The title mentions “from anecdotal evidence to experiment,” but neither the abstract nor the main text clearly references the specific anecdotal evidence or how it is compared to experimental results. It is recommended to supplement relevant background in the introduction or discussion section to enhance coherence between the title and the content.

2. The study includes only male runners, without involving female runners or other populations (e.g., professional athletes or different age groups). It is advised to explicitly discuss gender and population limitations in the limitations section and suggest expanding sample diversity in future research.

3. The study used only one minimalist shoe and one AFT shoe, which may not represent the diversity of similar footwear. It is recommended to emphasize this limitation in the discussion section and suggest including more brands or models in future studies.

4. The study relies on semi-structured interviews and subjective feedback, which may introduce recall bias or social desirability bias. It is suggested to clarify in the methods section how these biases were mitigated (e.g., through real-time recording or anonymous feedback).

5. Each trial involved only a 1.5 km run, which may not reflect real experiences during long-distance running or daily training. It is advised to acknowledge this limitation in the discussion and recommend extending the running distance or increasing the number of trials in future studies.

6. Although a six-phase thematic analysis was mentioned, there was no detailed explanation of how coding reliability and validity were ensured (e.g., inter-coder reliability checks). It is suggested to add specific steps or cite relevant methodological literature.

7. The results section is relatively long, but the discussion lacks in-depth analysis of certain findings (e.g., the conflict between “familiarity” and “novelty”). It is suggested to further explore the mechanisms or theoretical implications of these phenomena by referencing existing literature.

8. The terms “AFT” and “VP4” are used interchangeably (e.g., in tables), which may cause confusion. It is recommended to consistently use either “AFT” or the full name throughout the text.

9. Although ethical approval was mentioned, it was not specified whether participants signed written consent forms or how their privacy was protected. It is advised to provide these details in the methods section.

10. The conclusion mentions “gradual adaptation to new shoe types,” but does not provide specific transition plans (e.g., time, intensity). It is suggested to include practical recommendations based on the study results in the discussion or conclusion section.

11. It may be beneficial to compare these findings with the data-driven gait recognition approach proposed by Xu et al., to explore the complementarity between subjective feedback and objective identification. Such a comparison could enrich the understanding of how footwear choices influence gait and provide a theoretical foundation for future multidimensional analyses that integrate both qualitative and quantitative methods. (A new method proposed for realizing human gait pattern recognition: Inspirations for the application of sports and clinical gait analysis (https://doi.org/10.1016/j.gaitpost.2023.10.019))

**Do you want your identity to be public for this peer review?** For information about this choice, including consent withdrawal, please see our Privacy Policy

Reviewer #1: **Yes: ** Diana Ciubotariu

Reviewer #2: No

---

## [Author Response · Author response to Decision Letter 1]

16 Jul 2025

Response to Reviewer and Editor comments uploaded with submission

---

## [Decision Letter · Decision Letter 1]

9 Nov 2025

Dear Dr. Hébert-Losier,

Thank you for submitting your manuscript to PLOS ONE. After careful consideration, we feel that it has merit but does not fully meet PLOS ONE’s publication criteria as it currently stands. Therefore, we invite you to submit a revised version of the manuscript that addresses the points raised during the review process.

We look forward to receiving your revised manuscript.

Kind regards,

Seyed Hamed Mousavi

Academic Editor

PLOS ONE

Journal Requirements:

Reviewers' comments:

Reviewer's Responses to Questions

**Comments to the Author**

Reviewer #1: All comments have been addressed

Reviewer #3: All comments have been addressed

Reviewer #4: (No Response)

2. Is the manuscript technically sound, and do the data support the conclusions?

Reviewer #1: Yes

Reviewer #3: Yes

Reviewer #4: Yes

3. Has the statistical analysis been performed appropriately and rigorously?

Reviewer #1: Yes

Reviewer #3: Yes

Reviewer #4: N/A

4. Have the authors made all data underlying the findings in their manuscript fully available?

Reviewer #1: Yes

Reviewer #3: Yes

Reviewer #4: Yes

5. Is the manuscript presented in an intelligible fashion and written in standard English?

Reviewer #1: Yes

Reviewer #3: Yes

Reviewer #4: Yes

Reviewer #1: The authors have answered the comments of the reviewers. I have considered the article almost good in its initial version, so now it is adequate for publishing.

Reviewer #3: (No Response)

Reviewer #4: The manuscript presents a well-designed and engaging qualitative exploration of runners’ perceptions of minimalist and advanced footwear technology (AFT) shoes. The topic is timely and contributes meaningfully to the growing body of literature on subjective footwear experiences. It appears my review is after one round of peer reviews already occurred. Thus, I only have 1 major and a few minor concerns to address.

Major Concern

1. Study Design Classification

This study is primarily qualitative, with limited quantitative data (e.g., rankings, run times). As currently described, it is not clear that the study meets the criteria for a mixed-methods design. Please justify the “mixed-methods” terminology or modify the study design description to be qualitative.

Minor Concerns

Introduction

Third paragraph

-While it did not employ a Delphi approach, I consider Frederick (2022, Footwear Science, “Let’s Just Call it Advanced Footwear Technology”) to be the accepted definition for AFT. I recommend citing this paper as the field’s definitional reference.

-When discussing instability in AFT, consider also citing Hannigan et al. (2024, “Injury and performance-related running biomechanics in advanced footwear technology compared to minimalist footwear”), which demonstrated greater eversion excursion in AFT compared to minimalist footwear, supporting your claim. Citing Tenforde et al. (2023) in this section would also be warranted, as it is cited later but conceptually aligns here.

Fourth paragraph

-Minor point, but most AFT could be considered maximalist, so stating they are similar to traditional shoes (which generally lack a carbon plate, lack highly compliant foam, and have lower stack heights) appears a bit misleading

Fifth paragraph

-If possible, add a brief note explaining why only male runners were recruited, as this will help contextualize the study’s purpose and generalizability.

Methods

-Please explain whether the male-only design was chosen to limit variability (e.g., controlling for sex-based biomechanical differences) or due to recruitment convenience.

-Line 102 – “the running of the study” likely refers to the study’s conduct, not literal running alongside participants. Consider alternative wording such as “the conduct of the study” to avoid confusion.

-Line 123 – The description of your qualitative approach includes multiple philosophical terms (“interpretivist epistemology,” “constructivist ontology,” “neo-positivist methods”). While technically correct, the paragraph may be unnecessarily dense. The final sentence emphasizing your phenomenological approach conveys the key point effectively. Consider simplifying or consolidating this section for clarity.

-Please confirm whether participants wearing the Garmin 245 Music watches were not listening to music during the trials, as this could otherwise influence perceived comfort or performance.

Results / Quotations

-Several participant quotes contain minor grammatical or spelling errors. Because these were verbal interviews, please clarify your editorial policy on quote correction. If readability is affected, limited grammatical correction may be acceptable, provided meaning is preserved.

Line 218: “their” should be “they’re.”

Line 232: “the” should be “they are.”

Line 368: “your” should be “you’re.”

-Some participant quotes are in italics while some are in single quotation marks — should this be revised for consistency?

Results

-Line 403: Please clarify whether participants changed their responses from “would not buy” to “would buy,” or the reverse, when describing shifts in purchasing decisions.

**Do you want your identity to be public for this peer review?** For information about this choice, including consent withdrawal, please see our Privacy Policy

Reviewer #1: **Yes: ** Diana Ciubotariu

Reviewer #3: No

Reviewer #4: No

---

## [Author Response · Author response to Decision Letter 2]

12 Nov 2025

Details provided in attached document

---

## [Decision Letter · Decision Letter 2]

24 Nov 2025

Dear Dr. Hébert-Losier,

Thank you for submitting your manuscript to PLOS ONE. After careful consideration, we feel that it has merit but does not fully meet PLOS ONE’s publication criteria as it currently stands. Therefore, we invite you to submit a revised version of the manuscript that addresses the points raised during the review process.

**Dear Authors,**

We look forward to receiving your revised manuscript.

Kind regards,

Seyed Hamed Mousavi

Academic Editor

PLOS ONE

Journal Requirements:

Reviewers' comments:

Reviewer's Responses to Questions

**Comments to the Author**

Reviewer #4: All comments have been addressed

2. Is the manuscript technically sound, and do the data support the conclusions?

Reviewer #4: Yes

3. Has the statistical analysis been performed appropriately and rigorously?

Reviewer #4: N/A

4. Have the authors made all data underlying the findings in their manuscript fully available?

Reviewer #4: Yes

5. Is the manuscript presented in an intelligible fashion and written in standard English?

Reviewer #4: Yes

Reviewer #4: All comments were addressed and the paper is ready for publication.

I'll add one minor note. It appears the authors adjusted the discussion section in response to my comment below, while my comment was instead directed at the introduction section. I'll leave it to the authors whether to make any further edits to the introduction section, as the edits made did strengthen the discussion section.

Regarding my comment: "When discussing instability in AFT, consider also citing Hannigan et al. (2024, “Injury and performance-related running biomechanics in advanced footwear technology compared to minimalist footwear”), which demonstrated greater eversion excursion in AFT compared to minimalist footwear, supporting your claim. Citing Tenforde et al. (2023) in this section would also be warranted, as it is cited later but conceptually aligns here.

**Do you want your identity to be public for this peer review?** For information about this choice, including consent withdrawal, please see our Privacy Policy

Reviewer #4: No

---

## [Author Response · Author response to Decision Letter 3]

25 Nov 2025

The minor request has been applied, with response to the reviewer uploaded.

---

## [Editor Report · Decision Letter 3]

26 Nov 2025

How does it feel to run in minimalist and advanced footwear technology shoes: A qualitative study involving male recreational runners

PONE-D-25-14476R3

Dear Dr. Hébert-Losier,

We’re pleased to inform you that your manuscript has been judged scientifically suitable for publication and will be formally accepted for publication once it meets all outstanding technical requirements.

Kind regards,

Seyed Hamed Mousavi

Academic Editor

PLOS ONE
---

## [Editor Report · Acceptance letter]

PONE-D-25-14476R3

PLOS One

Dear Dr. Hébert-Losier,

I'm pleased to inform you that your manuscript has been deemed suitable for publication in PLOS One. Congratulations! Your manuscript is now being handed over to our production team.

Kind regards,

on behalf of

Dr. Seyed Hamed Mousavi

Academic Editor

PLOS One